# Winning Both the Accuracy of Floating Point Activation and the Simplicity of Integer Arithmetic

**Yulhwa Kim**[1], **Jaeyong Jang**[1], **Jehun Lee**[1], **Jihoon Park**[1], **Jeonghoon Kim**[2],
**Byeongwook Kim**[2], **Baeseong park**[2], **Se Jung Kwon**[2], **Dongsoo Lee**[2], **Jae-Joon Kim**[1]
[1]Seoul National University, [2]NAVER Cloud
{yulhwakim,jaeyongjang,jehun.lee,jihoonpark,kimjaejoon}@snu.ac.kr,
{jeonghoon.samuel,byeonguk.kim,baeseong.park,sejung.kwon,
 dongsoo.lee}@navercorp.com

## Abstract

Even though floating point (FP) numbers have been adopted as a de facto standard data format for deep learning computing, the complexity of FP arithmetic impedes a broader deployment of Deep Neural Networks (DNNs). Recent works such as quantization have attempted to replace the FP matrix multiplication (MatMul) of DNNs with simple integer MatMul by transforming the datatypes of both weights and activations into integers. Unfortunately, unlike weight values that are static, it is challenging to represent dynamic activations with integers. In this paper, to simultaneously achieve the accuracy of FP activation and the simplicity of integer arithmetic, we present a method for replacing FP arithmetic with integer one without changing FP activations in the storage format while weights are quantized. The proposed method pre-aligns the significands of FP activations just ahead of the MatMul on-the-fly so that the aligned significands (integers) can be used for the computation. Inspired by an observation that conventional FP arithmetic does not produce precise results due to rounding, we demonstrate that our proposed integer arithmetic-based scheme can produce the same level of errors as that of the FP arithmetic in case DNNs use FP activations and quantized weights. Experimental results show that the hardware based on the proposed scheme shows significant improvement over FP arithmetic-based designs in terms of energy efficiency and throughput-per-area while maintaining a similar level of accuracy.

## 1 Introduction

Deep Neural Networks (DNNs) usually use Floating-Point (FP) number systems to represent a wide range of weight and activation values. Such a comprehensive representation, however, demands high computational complexity and cost for FP matrix multiplication (MatMul) (Sze et al., 2017). On the other hand, integer (a.k.a fixed-point) arithmetic logic is much simpler while consuming less energy compared to FP counterpart (Jouppi et al., 2021). As such, the computational efficiency of DNNs can be enhanced by replacing FP arithmetic with integer one. Accordingly, quantization has been actively studied as a promising technique to support DNN computations with integer arithmetic, as it maps the input values of a (virtually) continuous domain (FP numbers) to the output values of a discrete set (integers) (Jacob et al., 2018). Note that even though several studies have successfully quantized weights and activations of some target DNNs with low-precision integer values (Li et al., 2021; Wu et al., 2022), quantization is still challenging for numerous DNNs. In particular, activation values are known to be more difficult to be quantized than the weight parameters because activations are dynamically generated during inference while the distribution of weights is static. The uncertainty of the distribution of dynamic activation values limits the ability to estimate proper quantization range (Choi et al., 2018). Such issues on activation quantization become even more serious when DNNs involve highly non-linear activation functions (e.g., GeLU) or modules that increase the variance of the activations (e.g., softmax and normalization layers) (Jeon et al., 2020). As a result, while the weight parameters can be successfully quantized even for generative mod-

Figure 1: An example of FP summation with (a) conventional FP computation and (b) proposed method. The precise summation is described in the box on top.

els (Xu et al., 2018; Bai et al., 2019; Jeon et al., 2022; Park et al., 2022; Kwon et al., 2022; Frantar et al., 2022) and extra-large models such as GPT-NeoX-20B (Chung et al., 2020; Yao et al., 2022), activation quantization usually relies on intensive quantization-aware training or sophisticated investigation algorithms such as dynamic min/max searching (Tao et al., 2022). Note that activation quantization is mandatory if integer arithmetic logic is involved for MatMul operations. Thus, to avoid such significant efforts to quantize complex DNNs (mainly due to activation quantization), recent neural processing units tend to employ FP arithmetic units even for inference process at the cost of increased energy and area (Jouppi et al., 2021).

To address the challenges discussed above, we propose a scheme that can achieve both the accuracy of FP activations and the simplicity of integer arithmetic. Our motivation stems from an observation that most multiplications can be removed once weights are quantized to be binary-coded (Jeon et al., 2020). Then, consecutive FP additions are mainly required to perform MatMul, and hence, we find conventional FP units can be much simplified. To be more specific, when processing the MatMul of DNNs, our proposed method first pre-aligns the significands of FP activations to be added. Correspondingly, FP activations can be reformatted into integer values and FP arithmetic units (FPUs) can be replaced with integer units during MatMul operations. A naive pre-alignment for accurate computation requires very high-resolution integer units for the computation, which negates the benefits of using integer units. Inspired by an observation that conventional FP arithmetic does not guarantee the exact results due to rounding errors (Wilkinson, 1994), we show that the same level of computational error can be obtained even when the pre-aligned significands are aggressively truncated. We then implement an integer-based FP arithmetic unit (iFPU) hardware for MatMul computation based on the proposed scheme. A comprehensive evaluation of the iFPU on various DNNs shows that the iFPU significantly improves energy efficiency and throughput-per-area over the conventional FPU-based MatMul engine while maintaining the neural network accuracy.

## 2 BACKGROUND

### 2.1 FLOATING-POINT ARITHMETIC AND ROUNDING ERROR

FP format represents a number as $(-1)^s \times (m) \times 2^{(e-bias)}$ which consists of sign ($s$), exponent ($e$), and significand (or mantissa, $m$) (Muller et al., 2018). Float32 assigns 1 bit for $s$ and 8 bits for $e$. Precision ($p$), the effective bit count of the significand, is 24 bits (among which 23 bits are explicitly stored). Bfloat16, which has been gaining popularity in the field of deep learning, intensely cuts down stored significand bits to 7 (compared to 23 in float32) to lower the total number of bits per value, and thereby reduces memory footprint (Wang & Kanwar, 2019). The bias of the exponent term is usually set to half of the exponent maximum.

FP format can cover a wide range of numbers by separating the significant digits and the scale of the number. Note that because of the precision limits, there is a gap between two consecutive FP numbers. Such a gap is called a unit of least precision ($ulp$) whose value is represented by the least significant digit. Hence, it is hard to represent real numbers precisely with FP format even if the numbers are in the dynamic range of the FP format, and rounding is required for converting real numbers into FP numbers. FP arithmetic typically normalizes significands for each computation, and the rounding operation is followed by the normalization to convert the computation result into an FP number. Round-to-nearest is the most frequently chosen as a rounding mode where the difference between the real value and the round-off value can be as large as half of $ulp$, and its relative error is bounded by $\epsilon = \frac{1}{2} ulp = 2^{-p}$, which is referred to as machine epsilon. Both $ulp$ and $\epsilon$ are widely used to evaluate the accuracy of numeric calculations (Goldberg, 1991).

As every FP operation includes the rounding stage, rounding error is unavoidable in FP arithmetic. Although the error of a single FP arithmetic operation may be small enough to be ignored, the error can be substantial if a series of multiple FP arithmetic results are accumulated. For example, an inner product of MatMul involves multiple FP additions in a row and the FP summation piles up the rounding error of each FP adder (Figure 1(a)). Accordingly, numerous solutions have been introduced to compensate for the error of the FP summation (Muller et al., 2018). Such error compensations cause an additional computation burden for tracking and fixing the error. Since the effect of the rounding errors on DNN accuracy is negligible, popular deep learning frameworks such as PyTorch and CuDNN (Paszke et al., 2019; Chetlur et al., 2014) allow the rounding errors (without the compensation algorithms) in favor of simple computation. Note that as the level of rounding error depends on the precision $p$ (only 8 bits for bfloat16), the error becomes noticeable for bfloat16. Therefore, summation of bfloat16 values uses float32 adders (instead of bfloat16 adders) to preserve the accuracy of accumulated results (Wang & Kanwar, 2019; Intel, 2018; Henry et al., 2019).

## 2.2 Related Works

**Block Floating Point (BFP)** has been proposed as a compromise between FP and integer formats. It assigns a single shared exponent to a group of FP values while maintaining individual significands (Wilkinson, 1994). The BFP has drawn attention as a flexible low-bit numeric format for quantization because the shared exponent can represent the dynamic range of values with little overhead. Hence, BFP can achieve a higher compression ratio than integer formats (Zhang et al., 2022a). In addition, since the individual significand values are integer, the BFP formats enable simpler computation than FP formats (Köster et al., 2017). Note that a critical limitation in previous works based on BFP formats is that the same level of accuracy as that of conventional FP computations cannot be guaranteed (even theoretically). Previous works tend to find the optimal BFP formats with the least memory/computation density by evaluating DNN accuracy for various bit resolution and group sizes (Song et al., 2018; Lian et al., 2019; Rouhani et al., 2020). Another drawback in some previous works on BFP is that DNNs with BFP format need to be fine-tuned usually by quantization-aware training to improve the accuracy (Zhang et al., 2022a; Rouhani et al., 2020). Since a quantized neural network allows only one fixed block size that is optimized for target hardware during training, a neural network needs to be retrained for different hardware choices if a block size differs.

**Truncated binary multipliers** with error compensation schemes have been proposed to reduce the number of outputs in integer multiplications (Petra et al., 2009). While both the truncated multipliers and our proposed work use the truncations to improve computational efficiency, there are critical differences between them. In the truncated binary integer multipliers, the amount of the truncated bits is fixed while it varies in FP additions cases which our work focuses on. In addition, (Petra et al., 2009) presents a truncation error correction function utilizing the fact that some of the truncated partial products share the same inputs with the remaining partial products, so they have correlations with the remaining partial product values. Unfortunately, in FP addition cases, the truncated significands do not have any correlation with the remaining bits so it is hard to devise similar error correction schemes. Hence, there is a strong need to develop alternative ways to control the truncation errors in FP operations.

## 3 Reconstruction of FP-based MatMul with integer arithmetic

### 3.1 Overview of the Proposed MatMul Reconstruction and Computation

In this section, we propose a methodology to reconstruct FP MatMul with integer arithmetic for efficient DNN computation, focusing on FP activations and quantized weights. In most cases, the weight matrix with $m$-bit quantization can be expressed as a binary-coded matrix: $\sum_{b=1}^{m} \alpha_b \cdot B_b$ where $\alpha_b$ is a scaling factor and $B_b$ is a binary weight matrix of each bitplane. Here, $\alpha_b$ can be a power of 2 for uniform quantization or can be an FP value for non-uniform quantization. MatMul is composed of multiple dot products, and a dot product between activations and weights is defined as $\sum_{k=1}^{n} (a_k \times w_k)$ ($a$: activation, $w$: weight, $n$: fan-in of the layer). If we apply binary-coded weights and properly change the order of the operations, we can rewrite the dot product as follows:

$$\sum_{b=1}^{m} \alpha_b \sum_{k=1}^{n} (a_k \times B_{b,k}), \; B_{b,k} \in [-1, +1] \tag{1}$$

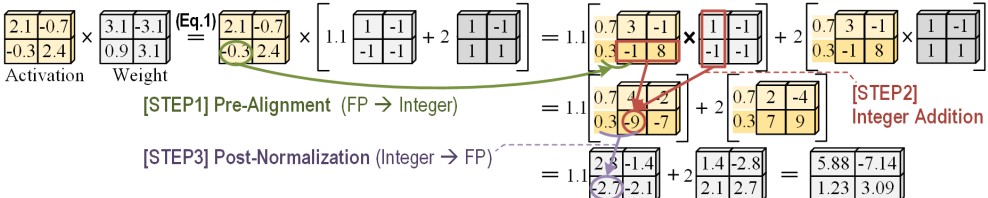

Figure 2: Overview of the proposed MatMul computing scheme for DNNs with FP activations.

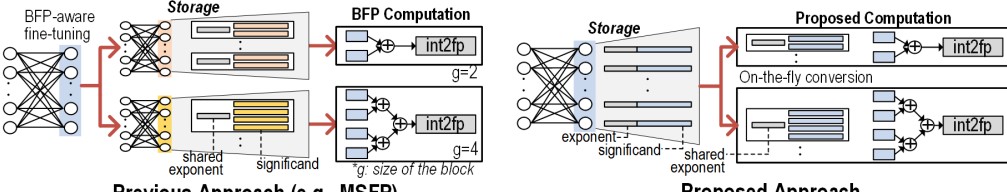

Figure 3: Comparison of a previous approach (e.g., MSFP (Rouhani et al., 2020)) and the proposed approach for applying block floating point (BFP) to DNN computation. In the case of MSFP, the original network needs to be retrained for the MatMul engines with different block sizes, but in the proposed scheme, the original network can be fed into the engines with any block sizes.

For each bitplane, MatMul of weights and activations is reconfigured as the addition/subtraction of activation values except for a few $\alpha_b$ multiplications that are necessary to merge the outputs from each bitplane. Because FP multiply-accumulate operations require more hardware resources than FP additions, even such a reconfiguration of matrix multiplication to remove most multiplications can improve the efficiency of DNN computations significantly (Jeon et al., 2020). Even so, because FP additions are still computationally more expensive than integer additions, replacing FP additions with integer additions can save even more energy and area. Therefore, we propose to reconstruct FP-based MatMul (Eq. 1) using integer additions (Figure 2). One of the key components of the proposed method is the pre-alignment, which reformats the FP activation values into integer values on-the-fly by sharing the exponent value among the activations that are fed to a dot product of the MatMul at a time. The pre-alignment finds the maximum of the exponents among the activations and aligns corresponding significands simultaneously based on the difference of each exponent and the maximum exponent. As a result, unlike conventional FP arithmetic that performs the alignment for each addition, our proposed computing methodology aligns the activation values once per MatMul, and thus, reduces the overall cost of the alignment process significantly. Note that as opposed to previous works that share the exponent among a block of inputs in the storage format (e.g., MSFP (Rouhani et al., 2020)), our design performs the exponent sharing during the computation. Since different exponents are allowed in the storage format in our scheme, we keep the representation power of the conventional FP format (Figure 3). Because pre-aligned activations can be represented by the aligned significands which are integer values, an FP addition of the MatMul can be replaced by an integer addition. After the whole summation process, the proposed method reformats the summation results back to FP values by normalizing the results with the maximum exponent found in the pre-alignment stage. Then, the computation results from each weight bitplane are multiplied by $\alpha_b$ and merged to finish the target MatMul operation.

As the exponent of float32 (or bfloat16) is 8-bit, the maximum amount of the significand shifting is 255 and the resolution of the aligned activation becomes 279 (or 263) bits. Note that such a large bit width might negate the benefits of using integer units. For example, while 32-bit integer addition consumes 10.3% energy of float32 addition, 279-bit integer requires a level of energy per addition comparable to that of float32 addition (Appendix B.1). To avoid the large design overhead, we propose to use only the top $t(= p + \delta)$ bits of the aligned activation when $\delta$ indicates the number of extra significand bits for reducing truncation error. Since the conventional FP addition also experiences errors due to truncation of significand, relatively small extra $\delta$ bits for the proposed method can derive a level of errors similar to that of conventional FP addition (as described in Figure 1).

## 3.2 COMPUTATION ERROR AFTER SIGNIFICAND TRUNCATION

To study the characteristics of errors in the proposed method with truncated significands, we first analyze the computation error with a single addition/subtraction between two FP values $x$ and $y$.

We assume $x > y \geq 0$, $x = x_0.x_1 \cdots x_{p-1}$, and $y = y_0.y_1 \cdots y_{p-1} \times 2^{-k}$ $(k \geq 0)$ without loss of generality, because only the difference between the exponents decides the amount of shifting and truncation. Here, $x_i$ and $y_i$ denote the binary value of $i$-th significand bit, and the leading bit $x_0$ is 1 for $x$ when $x > 0$. When either $k$ or $y$ is 0, there is no need for significand shifting and truncation, and hence, integer-based FP arithmetic can guarantee the precise computation without any extra bit (i.e., $\delta = 0$).

When $k > 0$, we need to shift and truncate the significand of $y$ for the computation. For the alignment, $y$ should be shifted to right by $k$, so $y$ can be rewritten as $y = 0.0 \cdots 0y'_k y'_{k+1} \cdots y'_{k+p-1}$ where $y'_{k+i}$ is equal to $y_i$. As only the top $t(= p + \delta)$ bits of the significand remain after the truncation, the truncated result becomes $\bar{y} = 0.0 \cdots y'_k \cdots y_{t-1}$. When $\delta \geq k$, the difference between $y$ and $\bar{y}$ is 0. Otherwise, the difference between $y$ and $\bar{y}$ is bounded as follows:

$$|y - \bar{y}| = 0.0 \cdots 0y'_t \cdots y'_{k+p-1} \leq 2^{-(p+\delta-1)}(1 - 2^{-(k-\delta)}). \tag{2}$$

The relative error of the addition with the truncated significand is defined as follows:

$$e_{add} = \frac{|(x+y) - (x+\bar{y})|}{|x+y|} = \frac{|y - \bar{y}|}{|x+y|}. \tag{3}$$

By applying both $|x + y| \geq |x| \geq 1$ and Eq. 2 to Eq. 3, we can obtain

$$e_{add} \leq 2^{-(p+\delta-1)}(1 - 2^{-(k-\delta)}) \leq 2^{-(p+\delta-1)}. \tag{4}$$

Because the machine epsilon is given as $\epsilon = 2^{-p}$, $e_{add} \leq \epsilon$ when $\delta$ is 1 and $e_{add} \leq \frac{1}{2}\epsilon$ when $\delta$ is 2.

For subtraction, the relative error is defined similarly as follows:

$$e_{sub} = \frac{|(x-y) - (x-\bar{y})|}{|x-y|} = \frac{|y - \bar{y}|}{|x-y|}. \tag{5}$$

When $\delta \geq k$, $|y - \bar{y}|$ is 0 so that $e_{sub}$ is 0. The minimum of $x$ is 1, and $y$ has the maximum value when all $y'_{k+i}$s are 1. Correspondingly, $|x - y|$ is bounded as follows:

$$|x - y| \geq \begin{cases} 1 - 0.11 \cdots 1 = 2^{-p}, & \text{for } k = 1 \\ 1 - 0.0 \cdots 01 \cdots 1 \geq 2^{-1} + 2^{-2} + \cdots + 2^{-(k-1)}, & \text{for } k \geq 2 \end{cases} \tag{6}$$

When $k$ is 1 and $\delta$ is 0, we get $|y - \bar{y}| \leq 2^{-p}$ from Eq. 2. For such a case, according to Eq. 6 and Eq. 5, we have $e_{sub} \leq 1$. The worst case happens when $x = 1$ and $y = 0.111 \cdots 1$. When $k \geq 2$, by applying Eq. 2 and Eq. 6 to Eq. 5, we get $e_{sub} \leq \epsilon$ for $\delta = 1$, and $e_{sub} \leq 1/2\epsilon$ for $\delta = 2$. As a result, regardless of FP formats, the proposed method has the error level as summarized in the following Remark 1.

**Remark 1** *The integer-based FP addition/subtraction has the same level of error as that of the conventional FP addition/subtraction with 1 extra bit, and the error becomes half with 2 extra bits.*

Note that the error of FP summation is the same as the accumulated value of errors from each addition (Muller et al., 2018). The reconstructed MatMul, however, induces an additional stage of

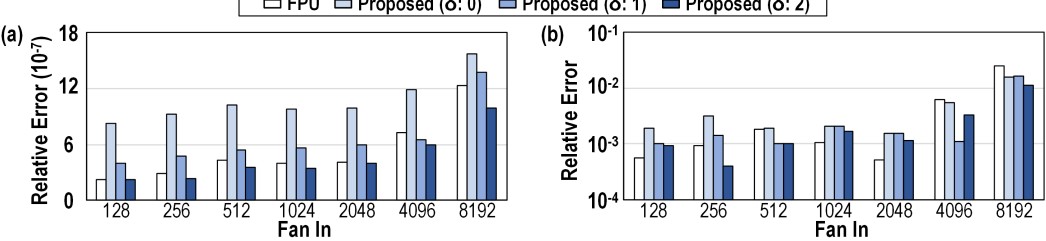

Figure 4: (a) Average and (b) maximum FP summation errors of conventional FP computation and the proposed method with extra bits ($\delta$=0,1,2) against the accurate FP summations with Schewchuk algorithm (Shewchuk, 1997).

converting integer summation results to FP values, and thus, additional rounding error during the FP formatting (Figure 5(a)). For example, to sum 128 FP values, a conventional FP-based MatMul has 127 error sources with bound $\epsilon$ while the reconstructed MatMul with 1 extra bit has 128 error sources with bound $\epsilon$ such that the reconstructed MatMul might experience a slightly larger error than conventional FP-based MatMul. Therefore, to guarantee the same error level as that of the conventional FP arithmetic, 2 extra bits are used for pre-alignment. Then, reconstructed MatMul has 127 error sources with bound $0.5\epsilon$ and an additional error source with bound $\epsilon$.

To verify the computation error of the proposed method, we randomly sample float32 values and compare the computation error of FP summation between conventional FP computation and the proposed method. To explore a wide range of float32 values, we sample $s$, $e$, and $m$ values independently assuming a uniform distribution, and then concatenate those values. We vary the fan-in (i.e., the number of values to be accumulated) from 128 to 8192, and sample 50,000 sets of FP numbers for each fan-in selection. The Schewchuk algorithm is employed to obtain accurate FP summation baseline data for error measurement (Shewchuk, 1997).

As shown in Figure 4, the proposed method produces a similar level of errors to that of the conventional FP arithmetic for various fan-in values when $\delta$=2. Because larger errors are more likely to be accumulated with larger fan-in, we see that both average and maximum errors tend to grow as the fan-in increases (Figure 4). Nonetheless, the average error ($12.3 \times 10^{-7}$) and the maximum error ($2.4 \times 10^{-2}$ or 2.4%) are relatively small even with 8192 fan-in, which justifies the current practice of implementing conventional FP additions without error correction for DNN inference. Correspondingly, the proposed method can support as precise numerical computation as conventional FP arithmetic does.

# 4 EXPERIMENT

## 4.1 IFPU: A MATMUL ENGINE FOR THE PROPOSED METHOD

**Overall Architecture.** To evaluate the proposed method with real hardware implementation, we first design a MatMul engine called iFPU. Figure 5 shows the overview of systolic iFPU architecture which adopts the design principle of Google's TPU (Jouppi et al., 2017). iFPU performs FP MatMul in the form of a set of FP summation (Eq. 1) that is physically implemented as integer summation for high efficiency. After the computation, the iFPU converts integer results into FP values through the int2fp converter at the end of the Processing Element (PE) arrays. Then, scale & accumulator is used to multiply $\alpha_b$ and add summation results of each weight bit-

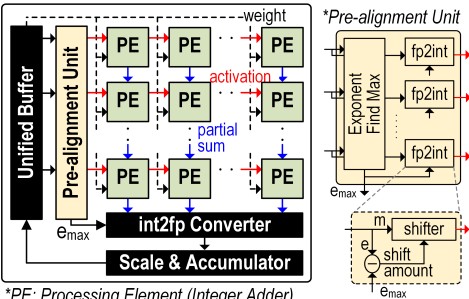

Figure 5: A block diagram of iFPU

plane to finish the MatMul (Eq. 1). The size of MatMul that can be processed in the iFPU at a time is bounded by the number of PEs, and as a practical design, we evaluate the iFPU with 32×32, 64×64, or 128×128 PEs for the experiment. When fan-in of the DNN layer exceeds the row count of PEs, activations of the layer are tiled to fit the row-count limit, and each tile is fed into the iFPU at a time and processed with integer adders in the PEs. To complete the entire MatMul, the computing results for different tiles should be merged, and for this, float32 adders (accumulator) are used again.

**Precision of Integer Adder.** As the PE array of the iFPU accumulates the pre-aligned and truncated significands, the size of the integer adder in each PE depends on $t$, which is determined by the precision of the given FP format ($p$) and extra bits ($\delta$) attached to control truncation error. Based on the theoretical analysis given in Section 3.2, the iFPU for float32 activations conducts 26-bit integer addition with $\delta = 2$. Though the iFPU introduces additional FP accumulations due to the MatMul tiling, the error level of integer-based FP addition with $\delta = 2$ is half of the conventional FP addition according to Remark 1. Therefore, the iFPU with $\delta = 2$ can still preserve the same level of computing error as that of conventional FP MatMul (Figure 6(a)). Furthermore, the iFPU for bfloat16 activations can be designed to be even smaller and more energy efficient by using smaller precision integer adders thanks to the reduced bit precision for significands. Interestingly, conventional bfloat16 accumulation still uses float32 adders to preserve the accuracy of accumulated

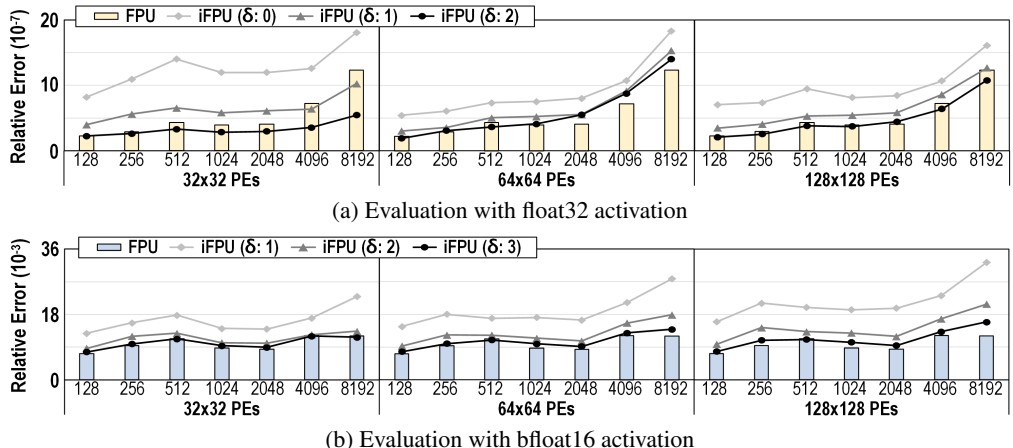

Figure 6: Numerical computation errors of MatMul for DNNs with FP activation. We measure the computation error of conventional FPU-based engine and the proposed iFPU against the accurate FP computation with Schewchuk algorithm (Shewchuk, 1997). The number of PEs and fan-in are annotated along the horizontal axis.

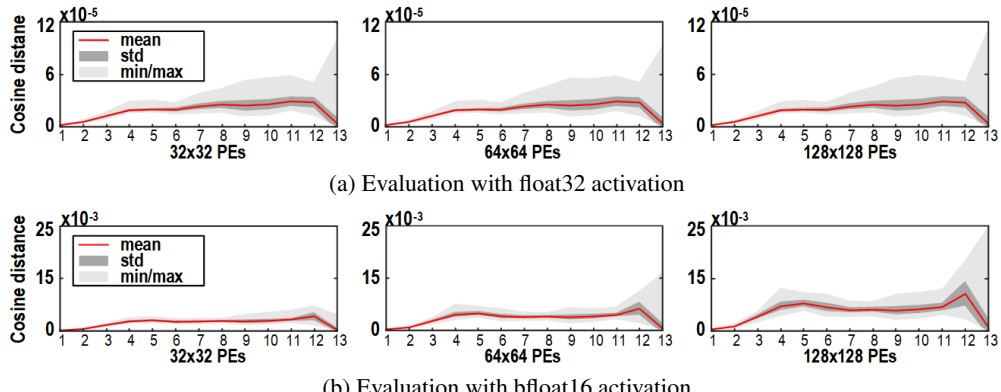

Figure 7: Cosine distance between MatMul results of BERT-base (task: MRPC) extracted from inference results using conventional FPU-based engine (NVIDIA RTX3090) and the proposed iFPU. The last feed-forward layer in each encoder block (1-12th layers) and pooler (13th layer) is used for the evaluation. The number of PEs and layer indices are annotated along the horizontal axis.

results (Wang & Kanwar, 2019; Intel, 2018; Henry et al., 2019). However, as the accumulated results are converted back to bfloat16, it is possible to maintain the accuracy of bfloat16 accumulation with less accurate adders than float32 adders. Figure 6(b) shows that the proposed bfloat16 iFPU with $\delta = 3$ (which uses 11 bit adders) provides comparable accuracy to that of conventional bfloat16 adders.

## 4.2 ANALYSIS OF THE DNN COMPUTATION ACCURACY

**MatMuls of DNN with iFPU vs FPU.** In the previous section, we compared the accuracy of the proposed integer-based FP MatMul with precise results. Since our goal is to replace the FPU with the proposed iFPU, it is also important to compare the computational difference between the conventional error-prone FPU-based engine and the iFPU. For an in-depth understanding of DNN inference with the iFPU, we first compare the inference output of each layer in the BERT-base model (Devlin et al., 2018) computed with an FPU-based engine (NVIDIA RTX3090) and the proposed iFPU. BERT-base uses 4-bit weight values and the target task is MRPC. In iFPU, MatMuls between weights and activations are processed with the proposed integer-based approach, but other operations such as softmax are processed by using conventional FPU. We employ cosine distance as the metric to measure the difference in layer outputs. Note that the cosine distance is 0 for two identical vectors and 2 for entirely opposite vectors. In this experiment, the last feed-forward layer in each

Table 1: Accuracy of DNNs inference with conventional FPU-based engine (NVIDIA RTX3090) and proposed iFPUs(-#rows/columns of PE arrays). The numbers in parentheses represent accuracy difference between FPU & iFPU.

| | float32 activation | | | bfloat16 activation | | |
|---|---|---|---|---|---|---|
| | VGG-9 | ResNet-18 | OPT-1.3B | VGG-9 | ResNet-18 | OPT-1.3B |
| FPU | 92.91 | 70.27 | 12.96 | 92.91 | 70.28 | 12.96 |
| iFPU-32 | 92.91 (+0.00) | 70.27 (+0.00) | 12.96 (+0.00) | 92.91 (+0.00) | 70.26 (-0.02) | 12.96 (+0.00) |
| iFPU-64 | 92.91 (+0.00) | 70.27 (+0.00) | 12.96 (+0.00) | 92.90 (-0.01) | 70.27 (-0.01) | 12.97 (+0.01) |
| iFPU-128 | 92.91 (+0.00) | 70.27 (+0.00) | 12.96 (+0.00) | 92.92 (+0.01) | 70.26 (-0.02) | 12.98 (+0.02) |
| | ResNet-50 | RegNet | MnasNet | ResNet-50 | RegNet | MnasNet |
| FPU | 76.32 | 78.18 | 75.99 | 76.33 | 78.17 | 75.96 |
| iFPU-32 | 76.31 (-0.01) | 78.18 (+0.00) | 75.99 (+0.00) | 76.38 (+0.05) | 78.18 (+0.01) | 75.97 (+0.01) |
| iFPU-64 | 76.31 (-0.01) | 78.18 (+0.00) | 75.99 (+0.00) | 76.38 (+0.05) | 78.18 (+0.01) | 75.96 (+0.00) |
| iFPU-128 | 76.31 (-0.01) | 78.18 (+0.00) | 75.99 (+0.00) | 76.40 (+0.07) | 78.18 (+0.01) | 75.97 (+0.01) |

| BERT--BASE w/ float32 activation | | | | | | | | |
|---|---|---|---|---|---|---|---|---|
| | CoLA | MRPC | SST-2 | STS-B | QQP | MNLI-m/mm | QNLI | RTE | Avg. |
| FPU | 56.36 | 89.05 | 91.51 | 87.52 | 83.73 | 81.95/82.56 | 89.00 | 70.04 | 81.28 |
| iFPU-32 | 56.36 | 89.05 | 91.51 | 87.52 | 83.73 | 81.95/82.56 | 89.00 | 70.04 | 81.28 (+0.00) |
| iFPU-64 | 56.36 | 89.05 | 91.51 | 87.52 | 83.73 | 81.95/82.56 | 89.00 | 70.04 | 81.28 (+0.00) |
| iFPU-128 | 56.36 | 89.05 | 91.51 | 87.52 | 83.73 | 81.95/82.56 | 89.00 | 70.04 | 81.28 (+0.00) |

| BERT--BASE w/ bfloat16 activation | | | | | | | | |
|---|---|---|---|---|---|---|---|---|
| | CoLA | MRPC | SST-2 | STS-B | QQP | MNLI-m/mm | QNLI | RTE | Avg. |
| FPU | 56.08 | 89.05 | 91.51 | 87.52 | 83.74 | 81.97/82.57 | 89.00 | 70.04 | 81.30 |
| iFPU-32 | 56.10 | 89.05 | 91.51 | 87.52 | 83.72 | 81.94/82.55 | 89.05 | 70.04 | 81.28 (-0.02) |
| iFPU-64 | 56.36 | 89.05 | 91.63 | 87.52 | 83.72 | 81.93/82.56 | 89.00 | 70.04 | 81.31 (+0.01) |
| iFPU-128 | 56.10 | 88.83 | 91.63 | 87.52 | 83.72 | 81.96/82.54 | 89.05 | 70.04 | 81.27 (-0.03) |

encoder block and pooler is chosen for evaluation. Figure 7 shows that the FPU and the iFPU produce almost identical outputs for each layer. The averages of the distance are less than $1.2 \times 10^{-6}$ and $2.5 \times 10^{-4}$ for float32 and bfloat16 activations, respectively. Moreover, the distance between layer outputs from the two engines remains close throughout the forward path. As a result, we can expect that the proposed iFPU can support DNN inference with almost the same accuracy as that of conventional FPU.

**DNN Inference Accuracy.** We select 7 types of DNN models to compare DNN model accuracy between the FPU and iFPU: BERT-base, VGG-9, ResNet-18, ResNet-50, RegNet-3.2GF, MnasNet-2.0, and OPT-1.3B. The accuracy of BERT-base is evaluated on the General Language Understanding Evaluation (GLUE) benchmark (Wang et al., 2019). VGG-9 (Simonyan & Zisserman, 2014) is evaluated on CIFAR-10 (Krizhevsky et al., 2009). ResNet-18, ResNet-50 (He et al., 2016), RegNet-3.2GF (Radosavovic et al., 2020), and MnasNet-2.0 (Tan et al., 2019) measure top-1 accuracy on ImageNet (Russakovsky et al., 2015). OPT-1.3B (Zhang et al., 2022b) is an open-sourced NLP model provided by Meta AI roughly matching the performance and sizes of the GPT-3 class of models and is evaluated by estimating the perplexity on WikiText-2 dataset (Merity et al., 2016). All DNN models use 4-bit weight values that are quantized by a binary-coding quantization scheme. Note that no modifications to DNN structures are needed to deploy the weight-quantized DNNs to various iFPUs because 1) activations are FP values and 2) iFPUs are designed to process any MatMul for DNNs as long as weights are quantized. Table 1 summarizes the DNN inference results. Because the iFPU can produce almost identical MatMul results as FPU, the proposed iFPUs preserve the DNN accuracy for both float32 and bfloat16 activations as we expected.

## 4.3 ANALYSIS OF COMPUTATION EFFICIENCY

**Setup.** To evaluate the efficiency of proposed iFPUs, we synthesize the proposed hardware in a 28nm CMOS technology. For a fair evaluation of the impact of replacing FP MatMul with integer-based MatMul, we also design two 'baseline' engines for the conventional FP-based MatMul (Fig-

ure 8). As the first baseline (FP-MAC), Figure 8(a) is designed with FP MAC units to process FP MatMul as a naive approach. In addition, as the second baseline (FP-ADD), Figure 8(b) is designed with FP adders to process FP MatMul reconfigured as Eq. 1. Because bitplanes of weight values are decomposed for FP-ADD and iFPU, binary weights are processed in a bit-parallel manner in FP-ADD and iFPU, while FP-MAC processes the whole weight values in each MAC unit. Compared to those two baseline engines, iFPU exhibits the lighter PEs along with additional units such as the pre-alignment unit and int2fp converter. Lastly, an int8 MatMul engine (INT8) is also implemented for the comparison between the proposed iFPU MatMul and integer MatMul.

**Results.** Simulation results using the synthesized hardware demonstrate that the proposed iFPUs can improve both energy and area compared to the baselines, as the FP units of the baseline engines are replaced with the more area/energy efficient integer units (Figure 9). For float32 activations, the proposed iFPU improves throughput-per-area (TOPS/$mm^2$) by up to $7.9\times$ and energy efficiency (TOPS/W) by up to $6.4\times$ compared to the FP-MAC baseline. For bfloat16 activations, the proposed iFPU achieves

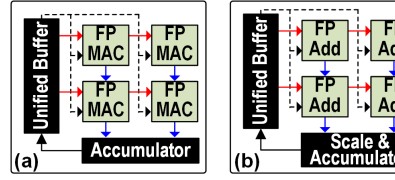

Figure 8: Baseline MatMul engines (a) FP-MAC and (b) FP-ADD

even larger improvements because the size of the corresponding integer-based unit is reduced as the bit resolution of the aligned-truncated significands is reduced by 15 bits compared to float32 activation cases. The throughput-per-area of the iFPU is improved by up to $9.9\times$ and energy efficiency is enhanced by up to $11.9\times$ compared to the FP-MAC baseline. The improvement over the baseline becomes larger as the number of PEs increases because the overhead of additional logic such as pre-alignment units in the proposed scheme can be amortized (detailed in Appendix C.2). We also compare the iFPUs with the INT8 engine. While bfloat16 activations close the gap between the FP-MAC baseline and the INT8 engine significantly in terms of throughput-per-area, iFPU (with bfloat16 activations) achieves even higher energy efficiency than the INT8 engine in some cases (Figure 9).

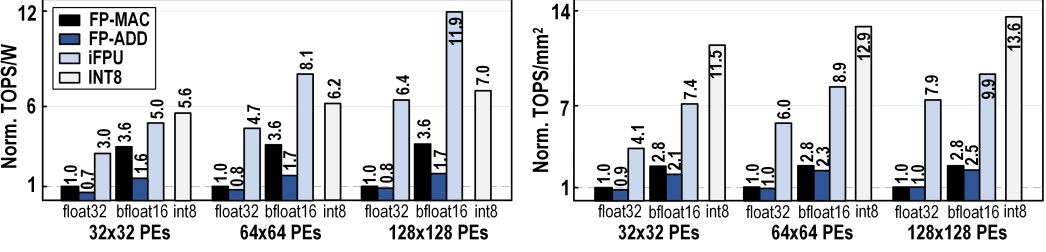

Figure 9: Normalized energy efficiency (TOPS/W) (left) and throughput-per-area (TOPS/$mm^2$) (right) of MatMul Engines: baselines and iFPUs for FP MatMul; INT8 for int8 MatMul. The number of PEs and target activation types are annotated along the horizontal axis.

## 5 CONCLUSION

The need to accomplish computing MatMul by using FP activations and quantized weights is increasing due to the growing usage of complex non-linear activation functions in DNN models such as Transformers. Conventional computing platforms such as CPU, GPU, and NPU, however, are inefficient in performing such computations. In this paper, we propose a new MatMul computing scheme dedicated to DNNs with FP activations and binary-coding weight quantization. The proposed method accelerates the FP MatMul of DNNs using the shared exponent and the integer arithmetic to improve computational efficiency. Previous works which also used the block floating point number with shared exponent often claim the validity of their design by presenting comparable DNN accuracy without verifying the robustness of MatMul results in a rigorous manner. We theoretically prove that the proposed scheme can produce the same error level as that of conventional FP arithmetic. To evaluate the computational efficiency of the proposed method, we design and synthesize a MatMul engine, iFPU, following the principle of integer-based operations. Experimental results support our claim that, compared to the conventional FPU-based design, the iFPUs accelerate the weight-only quantized DNNs with $6.4\times$ and $7.9\times$ higher energy efficiency and throughput-per-area for float32 activations, respectively. In addition, the iFPUs yield $11.9\times$ and $9.9\times$ higher energy efficiency and throughput-per-area, respectively, when associated with bfloat16 activations.

ACKNOWLEDGMENTS

This work was supported in part by Institute of Information communications Technology Planning Evaluation (IITP) grant funded by the Korea government (MSIT) (No. 2021-0-01343, Artificial Intelligence Graduate School Program (Seoul National University) (10%), and No.2021-0-02068, Artificial Intelligence Innovation Hub (10%)).

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

## A    COMPUTATIONAL COST OF FP ARITHMETIC VS. INTEGER ARITHEMETIC

Table 2: Energy of computing units synthesized in a 28nm tech node (MAC: multiply-accumulate).

| | MAC | | Multiply | | | Add | |
|---|---|---|---|---|---|---|---|
| | float32 | int8 | float32 | int32 | int8 | float32 | int32 |
| Energy per Operation | 1.51 pJ | 0.08 pJ | 1.23 pJ | 0.94 | 0.06 pJ | 0.28 pJ | 0.03 pJ |
| Normalized Energy | **18.9×** | 1.0× | 20.5× | 15.7× | 1.0× | 9.3× | 1.0× |

To cover a wide range of numbers, FP format does not fix the location of the radix point (Goldberg, 1991). Hence, FP arithmetic needs to handle input and output values with different scaling factors, and the FP arithmetic units need to align and normalize significands before and after each computation, respectively. The alignment and normalization logics consist of barrel shifters that can shift a data word by a specified amount, and the cost of the barrel shifter far exceeds the cost of other arithmetic logics in terms of both energy and area, increasing the cost of FP computation (Horowitz, 2014). Hence, in general, integer arithmetic logic is much smaller and consumes less energy than FP counterpart.

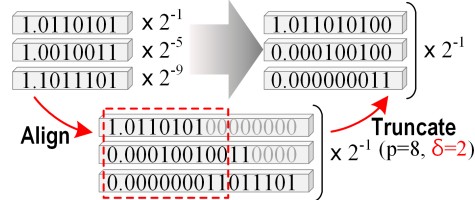

Figure 10: Area comparison of computing units (layouts synthesized in a 28nm node).

It is well known that 8-bit integer can achieve up to 4× throughput improvement compared to IEEE-754 single-precision format (float32) in widely used GPUs (Kim et al., 2021), as the throughput of 8-bit operations is generally 4× that of 32-bit operations (Harris, 2016). The advantage of integer can be magnified when the hardware platform moves to ASIC (Mishra et al., 2018). For in-depth understanding, we synthesize computing units for FP and integer in a 28nm tech node. As shown in Table 2, multiplication-accumulation (MAC) for float32 consumes 18.9× more energy than 8-bit integer (int8), a widely used integer format for quantized DNNs. Please note that a float32 MAC consists of a float32 multiplication and a float32 addition while an int8 MAC consists of an int8 multiplication and an int32 addition. The bit resolution of the adder for the int8 MAC is higher than that of the multiplier, because int8 multiplication results in 16-bit values and the bit resolution of MAC values increases as the number of accumulated values increases for integer format. In addition, the area cost of the integer unit is also much smaller than FP units as shown in Figure 10. Therefore, many studies have attempted activation quantization despite the various difficulties in the quantization process because both weight parameters and activations should be quantized to replace FP arithmetic with integer arithmetic.

## B    SUPPLEMENT FOR PROPOSED SIGNIFICAND TRUNCATION

### B.1    ENERGY IMPROVEMENT WITH SIGNIFICAND TRUNCATION

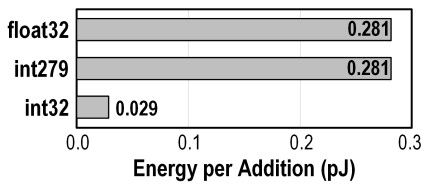
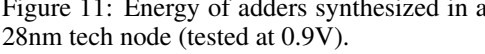

Figure 11: Energy of adders synthesized in a 28nm tech node (tested at 0.9V).

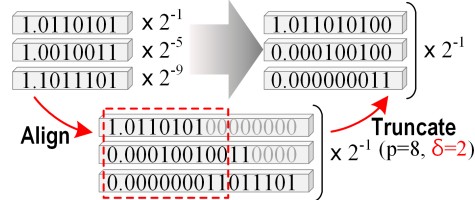

Figure 12: Example of the significand truncation followed by the pre-alignment.

With naive pre-alignment of float32 activations, the maximum amount of the significand shifting is 255 and the resolution of the aligned activation becomes 279 bits. As shown in Figure 11, while 32-bit integer consumes 0.029 pJ per addition, both float32 and 279-bit integer consumes 0.281 pJ

per addition. To avoid the large design overhead, we truncate the pre-aligned significands as shown in Figure 12. The aggressive truncation still did not cause accuracy degradation in FP additions as we described in the Section 3.2.

## B.2 Truncated Binary Multipliers vs. Proposed significand truncation

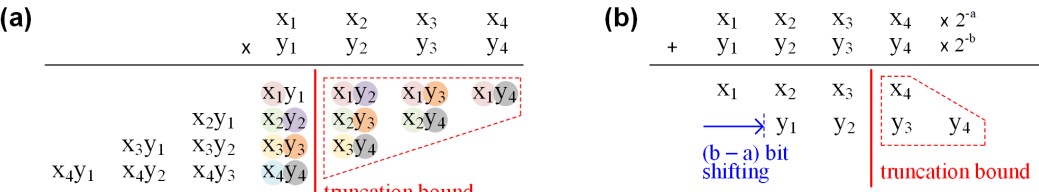

Figure 13: Comparison of the truncation scheme in the (a) truncated binary multiplier for integer multiplication and (b) proposed method for FP addition/subtraction.

Truncated binary multipliers (Petra et al., 2009) also discuss the truncation to improve computational efficiency, but there are critical differences between truncated binary multipliers and the proposed work as summarized in Figure 13. First of all, truncated binary multipliers deals with integer multiplications while the proposed work focuses on FP additions/subtractions. Due to the differences in the number format (integer vs. FP) and arithmetic operations (multiplications vs. additions/subtractions), the two works present completely different error analysis models and error reduction schemes.

The error analysis models between truncated binary multipliers and our case are different, because the amount of truncation is fixed in the truncated binary integer multipliers and the amount of truncation varies in our work as the amount of significand shift varies depending on the input data. Moreover, in truncated binary multipliers, the bit resolution of truncated output is defined by the application requirement. On the other hand, as we proposed to truncate the pre-aligned values to adopt lower-bit integers and improve computational efficiency, the proper bit resolution of truncated values should be found to meet the accuracy requirement in our case.

In addition, in integer multiplication case, some of the truncated partial products share the same inputs with the remaining partial products, so they have correlations with the remaining partial product values. (Petra et al., 2009) proposed an error minimization scheme which exploits such characteristics. On the other hand, in the FP addition/subtraction case, the truncated significands do not have any correlation with the remaining bits so it is hard to devise similar error correction schemes. Instead, we focused on the fact that conventional FP operation is also not precise due to the rounding of output significands so that we only need to match the error level of the proposed scheme to the conventional FP operations. Based on the facts, we showed a theoretical analysis such that the proposed integer-based FP addition/subtraction can have the similar error level as that of the conventional FP addition/subtraction when small number (1-2) of extra bits are attached to the shifted significands. With this finding, we can design an efficient integer-based FP addition logic without having complex error correction function estimated based on the truncated bits.

## C In-depth hardware analysis

### C.1 Detailed hardware description of the proposed iFPU

Figure 14 describes the proposed iFPU in detail. The proposed iFPU is a bit-flexible accelerator which can handle variable bitwidth of weight values. The iFPU processes weights in bit-parallel manner by processing each weight bitplane in different columns of the PE array. For example, 4-bit weights use 4 PE columns for the computation, and 8-bit weights use 8 PE columns for the computation. After the integer-based summations are done in each column of the PE array, the integer results are converted into FP values and multiplied by scaling factors which represent the significance of each bitplane. Then, computing results of each bitplane are merged in the accumulator (FP adder) to finish the MatMul. As the output resolution of FP accumulation remains the same regardless of the

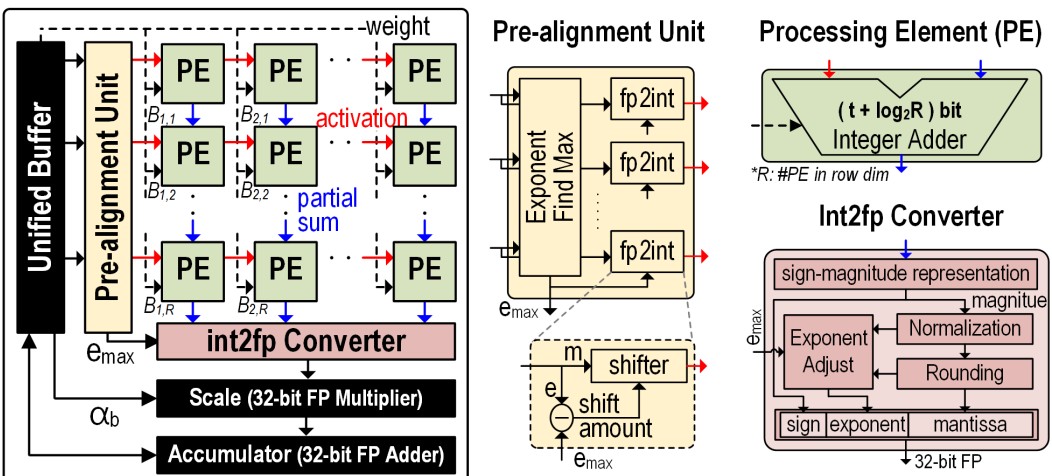

Figure 14: A detailed block diagram of iFPU. The iFPU processes weights in bit-parallel manner by processing each bitplane of the weights in each column of the PE array. ($B_{b,k}$: binary weights in Eq. 1)

size of the accumulation thanks to the characteristics of the FP format, the size of the accumulator does not need to increase for the increased weight bit width.

## C.2 AREA/ENERGY BREAKDOWN OF PROPOSED IFPU

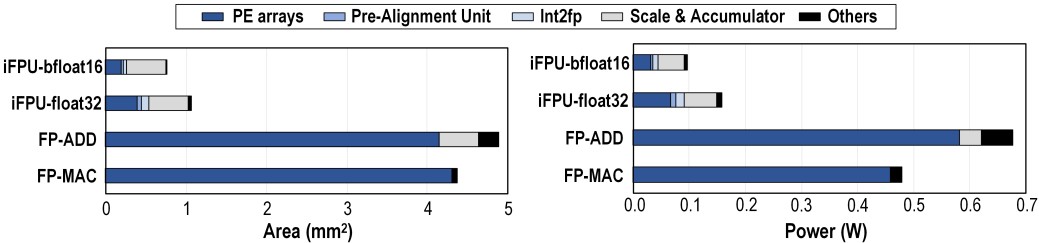

Figure 15: Area ($mm^2$) (left) and power (W) (right) of MatMul Engines: baselines and iFPUs for FP MatMul with 32x32 PEs.

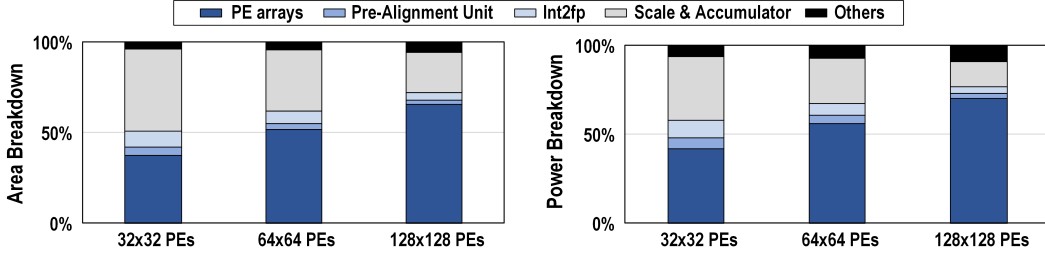

Figure 16: Area breakdown (left) and power breakdown (right) of proposed iFPUs with 32x32, 64x64, and 128x128 PEs.

In this section, the area and power of the MatMul engines designed in Section 4.3 are analyzed in more detail for deeper understanding of the proposed scheme. First, a breakdown of the area/power of various MatMul engines with 32x32 PEs is shown in Figure 15. FP-ADD reconstructs FP-MAC with a series of FP additions by separately processing each weight bitplane (Eq. 1), so to match the effective throughput of FP-ADD with that of FP-MAC in case of 4-bit weights, 4 FP-ADD

operations are used for the evaluation. Hence, though the area/energy of a single float32 adder is lower than that of a float32 MAC unit (Table 2), FP-ADD requires slightly larger area and power than FP-MAC. On the other hand, though iFPU also introduces $m$ times more operations than FP-MAC, iFPUs achieve large area and power reduction as the area/energy cost of PE arrays become significantly lower by replacing FP adders with integer adders. The area/power reduction is even larger in bfloat16 cases because smaller integer units can be used. As the area/power cost of PE arrays in iFPUs decreases, the relative portion of area/power of supporting logic (such as scale & accumulator) in the total area/power increases. Hence, the supporting logic accounts for more than half of the total area/power of iFPUs with 32x32 PEs. Meanwhile, the overhead of the supporting logic decreases as the size of PE arrays increases. We report the area/power breakdown of iFPUs with various number of PEs in Figure 16. The experimental results show that, as the size of PE arrays increase, the supporting logic is shared among more PEs and the overhead can be amortized.

### C.3  IMPACT OF THE WEIGHT BITWIDTH ON THE PROPOSED IFPU

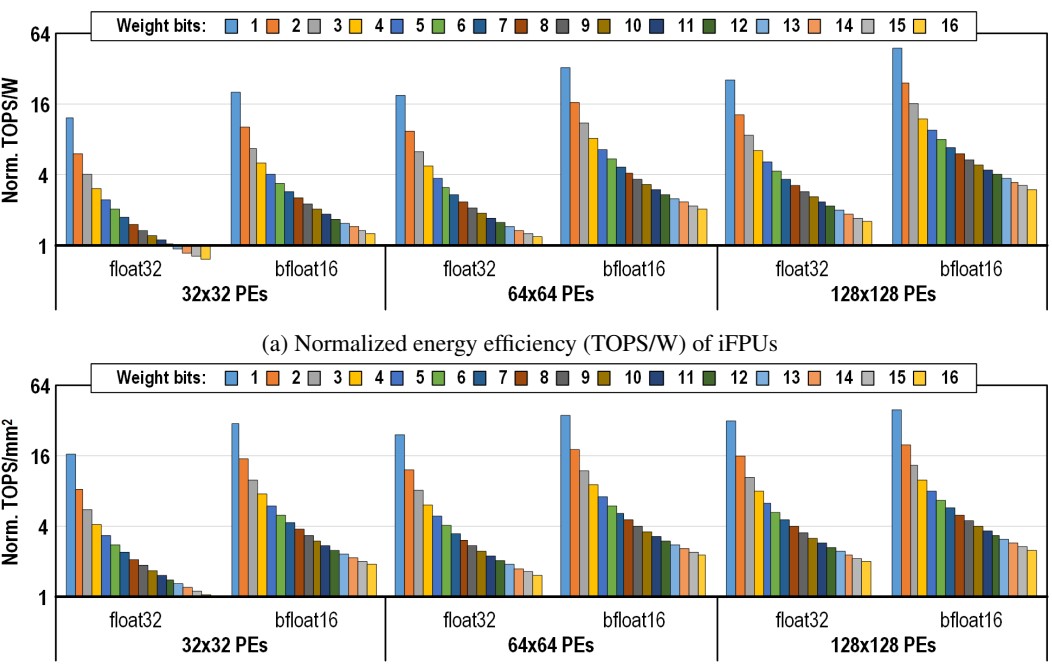

(a) Normalized energy efficiency (TOPS/W) of iFPUs

(b) Normalized throughput-per-area (TOPS/$mm^2$) of iFPUs

Figure 17: Computational efficiency of iFPUs normalized with that of the baseline FP MatMul engine (FP-MAC). Y-axis is the normalized value against FP-MAC and the iFPUs show higher efficiency than FP-MAC even for high-precision weight bits. The number of PEs and target activation types are annotated along the horizontal axis.

This section analyzes impact of weight bitwidth on the efficiency improvement achievable with the proposed iFPU. The experimental setup is the same as Section 4.3 except the weight bits. While only 4-bit weight cases are evaluated in Section 4.3, this section evaluates weights with 1 to 16 bits. Because the proposed scheme processes each bitplane of the weights in the bit-parallel manner, higher-bit weights require more operations with PE, scale, and accumulators. Hence, as shown in Figure 17, the benefits of the iFPUs diminish as the number of weight bits increases. Nevertheless, even for 8-bit weight case, iFPUs achieve better computational efficiency compared to the FP-MAC baseline.

### C.4  COMPARISON OF THE PROPOSED IFPU WITH INT4 MATMUL ENGINE

In Figure 18,, an int4 MatMul engine (INT4) is evaluated and compared with the other MatMul engines analyzed in Section 4.3. INT4 MatMul shows high energy efficiency and throughput-per-

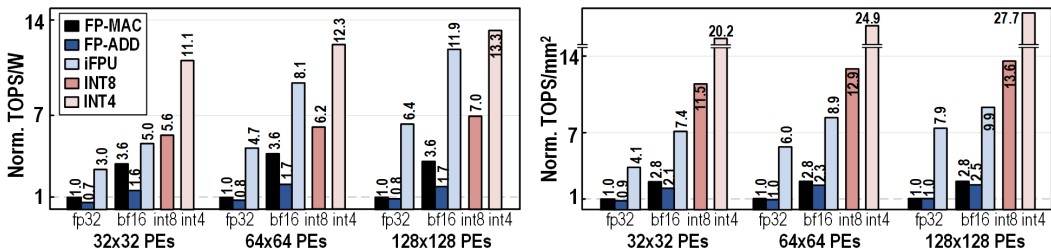

Figure 18: Normalized energy efficiency (TOPS/W) (left) and throughput-per-area (TOPS/$mm^2$) (right) of MatMul Engines: baselines and iFPUs for FP MatMul; INT8/INT4 for int8/int4 MatMul. The number of PEs and target activation types are annotated along the horizontal axis.

area. However, to take advantage of INT4 MatMul, both weight and activation should be quantized to 4 bits, which may not provide desired accuracy in many cases.

## C.5 Hardware evaluation with memory access

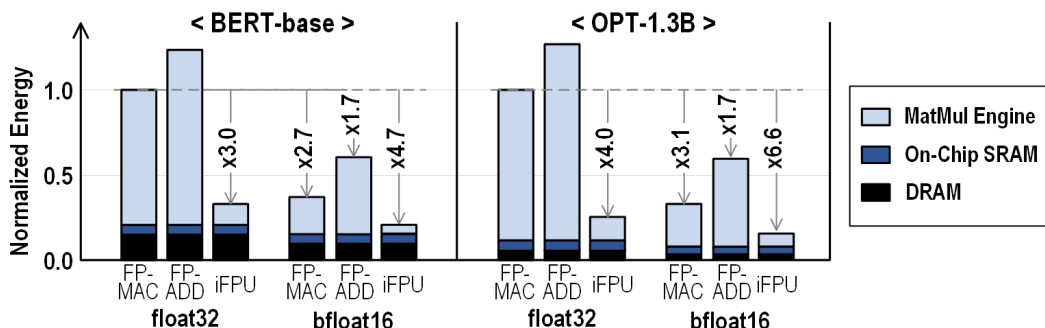

Figure 19: Normalized energy consumption of MatMul engines (FP-MAC, FP-ADD, and iFPU) with memory system. The inference energy is measured for BERT-based and OPT-1.3B with 4-bit weights and float32/bfloat16 activations.

**Setup.** To understand the effectiveness of the proposed method in the real computing scenario, the baselines (FP-MAC and FP-ADD) and the proposed iFPU with 128x128 PEs are further evaluated including memory access. For off-chip memory, we scaled down the bandwidth of HBM2 in TPU (Jouppi et al., 2021) considering the ratio of the number of PEs that make up Matrix Multiply Unit (MXU), which is 1:4 and adopted energy per bit of HBM2 from Table 2 in (Jouppi et al., 2021); we used the bandwidth of 153.5 GB/s and the energy per bit of 3.9 pJ/bit. We also scaled the size of the unified buffer (on-chip SRAM buffer) in (Jouppi et al., 2021) by dividing it by 4. The unified buffer size in our design was 32MB. For SRAMs, we used the 28nm CMOS memory compiler and the energy per bit of 0.155 pJ/bit was used. To overlap memory access with computation, double buffering scheme was adopted in the unified buffer.

**Results.** We evaluate a single batch inference of BERT-base and OPT-1.3B. We set the sequence length of BERT-base and OPT-1.3B as 128 and 1024 respectively. As double buffering hides the memory access latency, the proposed iFPU with memory model can achieve the same amount of throughput-per-area improvement as that of the baseline for the case in which memory access is not considered. On the other hand, the gain in energy efficiency slightly changes after considering memory access. As shown in Figure 19, the dram access energy accounts for a relatively small portion of total energy consumption in the baselines, because the data is intensively reused in the MatMul computation. As the proposed iFPU reduces the energy cost of computation, memory access energy becomes relatively significant in the proposed system. Thus, when considering the cost of memory access, the amount of improvement in the energy efficiency slightly decreases. Nevertheless, the iFPU with memory access still can improve the energy efficiency by up to 6.6x compared to FP-MAC baseline.

## D    FINE-TUNING CONDITION FOR BERT-BASE TRAINING

Table 3: Hyper-parameters for fine-tuning BERT-base on GLUE benchmark. The fine-tuning use AdamW optimizer and the number of training epochs is 10. The learning rates decay linearly and the weight decay is set to 0.01.

| Configuration | GLUE | | | | | | | |
|---|---|---|---|---|---|---|---|---|
| | CoLA | MRPC | SST-2 | STS-B | QQP | MNLI | QNLI | RTE |
| Batch size | 16 | 32 | 32 | 32 | 32 | 16 | 16 | 16 |
| Learning rate | 1e-4 | 1e-4 | 1e-4 2e-4 | 1e-4 | 5e-5 | 5e-5 | 5e-5 | 1e-4 |

