# OpenReview forum: "Winning Both the Accuracy of Floating Point Activation and the Simplicity of Integer Arithmetic"
_ICLR.cc/2023/Conference — ICLR 2023 poster_

### Official Review · Reviewer_yB16 · 2022-10-21

**Confidence:** 2
**Correctness:** 3
**Technical Novelty And Significance:** 3
**Empirical Novelty And Significance:** 3
**Recommendation:** 6

**Clarity, Quality, Novelty And Reproducibility:**

Overall, the paper is clear to understand.

However, Figure 1 would be more helpful if the example calculation was done on multiply and addition (as it appears in MatMul) to explain how the pre-alignment process works.

Questions:
The results on BERT-base seem inconsistent with the numbers presented in the original paper and on the huggingface model hub. Please elaborate.

**Strength And Weaknesses:**

# Strength
- The paper introduces an interesting approach for inferencing neural networks faster.
- The method only requires the weights to be quantized. This puts much less stress on the stability of the numerics and training compared to a fully quantized network where both the weights and the activations are quantized.
- Extensive experimental evaluation on NLP and CV tasks with comparison to fp32 and bf16 computations.

# Weaknesses
The paper misses an evaluation of important computer vision architectures. Particularly, the paper evaluates VGG9, which is undoubtedly outdated and not relevant anymore (only 92% on CIFAR, where 96% can be easily obtained with a ResNet and CUTOUT augmentation).
Moreover, the ResNet variant for ImageNet is rather small (ResNet18), whereas the "gold standard" for CV evaluations is the ResNet50. I believe larger models or new models (EfficientNet, ConvNext) would make the evaluation of the CV part more relevant for ICLR 2023.

The method seems to rely heavily on weight quantization, i.e., requiring the assumption that the network still performs well if the weights are expressed by low-bit integers.
It would be, therefore, relevant for the paper to provide results on how much the performance of a network suffers from the weight quantization.

**Summary Of The Paper:**

The paper proposed an efficient matrix multiplication hardware. Concretely, the paper performs MatMuls with quantized weights and floating-point activations operants and outputs a numerical approximation of the floating-point result.

**Summary Of The Review:**

Overall interesting approach with high potential real-world usage due to only weights required to be quantized.

---

> ### Author Response · Authors · 2022-11-15
> **Response to Reviewer yB16 (1/2)**
>
> Thanks for the constructive comment.
>
> We will now discuss 1) evaluation of the CV part with larger/new models, 2) impact of the weight quantization on DNN accuracy, 3) Figure 1 modification, and 4) results on BERT-base and its fine-tuning condition.
>
> **Response yB16-1. Evaluation of the CV part with larger/new models**
>
> Table yB16-A. DNN inference accuracy with conventional FPU and proposed iFPUs. (weight: int4, activation: float32)
> || ResNet-50 | MobileNetV2 | RegNet-3.2GF | MnasNet-2.0 |
> |---|:---:|:----:|:---:|:---:|
> |FPU		|	76.32	| 71.80	| 78.18	| 75.99 |
> |iFPU-32	|	76.31	| 71.80	| 78.18	| 75.99 |
> |iFPU-64	|	76.31	| 71.80	| 78.18	| 75.99 |
> |iFPU-128	|	76.31	| 71.80	| 78.18	| 75.99 |
>
> Table yB16-B. DNN inference accuracy with conventional FPU and proposed iFPUs. (weight: int4, activation: bfloat16)
> || ResNet-50 | MobileNetV2 | RegNet-3.2GF | MnasNet-2.0 |
> |---|:---:|:----:|:---:|:---:|
> |FPU		|	76.33	| 71.78	| 78.17	| 75.96 |
> |iFPU-32	|	76.38	| 71.75	| 78.18	| 75.97 |
> |iFPU-64	|	76.38	| 71.75	| 78.18	| 75.96 |
> |iFPU-128	|	76.40	| 71.74	| 78.18	| 75.97 |
>
> We evaluate the proposed method on larger/new CV models (ResNet50, MobileNetV2, MnasNet-2.0, RegNet-3.2GF) as shown in Table yB16-A and Table yB16-B. Similar to previous experimental results on smaller CV models, the proposed scheme did not experience accuracy degradation from FPU.
>
> As the proposed scheme is based on the weight-only quantized model, we used BRECQ [1], a PTQ framework, to obtain 4-bit weight models. Though the models in this experiment are not exactly the ones that the reviewer suggested, we believe that these models are large enough to make the experiment results more relevant. The experimental results (except MobileNetV2 results due to the space limit) are also updated on Table 1 of the revised paper.
>
> [1] Yuhang Li, et al. “Brecq: Pushing the limit of post-training quantization by block reconstruction.” In International Conference on Learning Representations, 2021.
>
> **Response yB16-2. Impact of the weight quantization on DNN accuracy**
>
> As the reviewer correctly pointed out, our efficient inference system needs to be supported by quantization methods that can achieve a small number of quantization bits for weights. Recent publications have presented numerous successful techniques and experimental results to lower the number of weight quantization bits while keeping the activation in FP format (please refer to 6 references below). For example (specifically for large-scale language models employing more than a few billions parameters), nuQmm[4] does not degrade zero-shot performance for the OPT models over 6.7B when quantized by using less than 4 bits. AlphaTuning [5] shows that even 1-bit quantization is available to perform the fine-tuning process. In addition, GPTQ [6] presented no accuracy degradation even when weights are (uniformly) quantized by using 3-4 bits.
>
> We believe that one of our major contributions is to provide a practical design methodology to enable weight-only quantization (while most previous studies include activation quantization steps as a mandatory procedure). Even though our work does not propose a new quantization technique, thanks to the reviewer’s constructive comments, we feel that it is necessary to include a thorough list of related works. Accordingly, in the Introduction of the revised manuscript, we added more relevant publications.
>
> [1] Chen Xu, et. al. “Alternating multi-bit quantization for recurrent neural networks.” In International Conference on Learning Representations, 2018.
>
> [2] Yu Bai, et. al. “Proxquant: Quantized neural networks via proximal operators.” International Conference on Learning Representations, 2019.
>
> [3] Yongkweon Jeon, et al. “Mr.biq: Post-training non-uniform quantization based on minimizing the reconstruction error.” In Proceedings of the IEEE/CVF Conference on Computer Vision and Pattern Recognition, pp. 12329–12338, 2022.
>
> [4] Gunho Park, et al. “nuqmm: Quantized matmul for efficient inference of large-scale generative language models.” arXiv preprint arXiv:2206.09557, 2022.
>
> [5] Se Jung Kwon, et al. “Alphatuning: Quantizationaware parameter-efficient adaptation of large-scale pre-trained language models.” arXiv preprint arXiv:2210.03858, 2022.
>
> [6] Elias Frantar, et al. “Gptq: Accurate post-training quantization for generative pre-trained transformers”. arXiv preprint arXiv:2210.17323, 2022.

---

> > ### Author Response · Authors · 2022-11-15
> > **Response to Reviewer yB16 (2/2)**
> >
> > **Response yB16-3. Figure 1 Modification**
> >
> > Thank you for the constructive comments. We are still reluctant to update Figure 1 as we believe that Figure 2 already shows how the pre-alignment works in the FP-based MatMul. Please check whether Figure 2 serves for better understanding and let us know if you have comments for further improvement. We would be happy to revise the figure.
> >
> > The key idea of the proposed work is to reconstruct FP sum with pre-alignment to improve the computational efficiency. When the weights are quantized, we can decompose each bitplane of weight values with Eq. 1. In this case, FP-based MatMul can be reconfigured as the set of FP sum as shown in Figure 2. Then, if we can improve the computational efficiency of FP sum, the overall efficiency of FP-based MatMul can be improved. Hence, we designed a method to reconstruct FP sum with pre-alignment/post-normalization. In Figure1, we intended to present the reason why the proposed FP sum with pre-alignment can preserve a similar level of computational accuracy to that of conventional FP sum.
> >
> > **Response yB16-4. Results on BERT-base and its fine-tuning condition**
> >
> > As the reviewer correctly pointed out, our results on BERT do not agree with the numbers presented in the original paper and on the huggingface model hub. Please note that the numbers do not match between the original paper and the official BERT example on the huggingface model hub. The results of BERT vary from report to report, as the accuracy highly depends on the fine-tuning conditions. Hence, we report the fine-tuning condition that we used in Table 3 of Appendix D for reproducibility.
> >
> > Thanks again for the comments.

---

### Official Review · Reviewer_7bGi · 2022-10-23

**Confidence:** 4
**Correctness:** 3
**Technical Novelty And Significance:** 3
**Empirical Novelty And Significance:** 2
**Recommendation:** 6

**Clarity, Quality, Novelty And Reproducibility:**

The proposed concept is presented clearly and is novel. Regarding reproducibility, please answer question 3 above.

**Strength And Weaknesses:**

The proposed idea is very interesting. However, I have several questions to the authors:

1) The proposed work focuses much on the compute efficiency. I wonder if this is really critical, as the memory access cost is known to usually be a bottleneck for DNN accelerator architecture. Can the authors comment on how significant reducing the cost of compute is when memory access is taken into account?

2) The authors claim that FP is needed to maintain accuracy. This is usually true when quantization methods do not have a good handle on the range vs resolution trade-off in the data. However, for a properly clipped quantizer, integer quantization can be shown to be as accurate as FP [1]. In such a case, integer arithmetic can naturally be used. Can the authors compare their work to such an fully-integer baseline?
[1] Sakr, Charbel, et al. "Optimal Clipping and Magnitude-aware Differentiation for Improved Quantization-aware Training." International Conference on Machine Learning. PMLR, 2022.

3) I am curious as to how the DNN accuracy number were obtained in the experimental section. Specifically, how is the FP to INT MatMul emulation performed? The implementation of CUDA kernels, which are invoked by deep learning frameworks, are fast because of the reduction operations that can eliminate the need to store intermediate results in memory. To emulate the proposed method, the authors need to perform an element-wise multiplication, followed by truncation, and then successive additions and truncations. This is very memory and time consuming. Do the authors have a trick to bypass these challenges in the emulation? Or are the experiments simply very slow?

**Summary Of The Paper:**

The paper proposes to use integer arithmetic on truncated floating-point operands. The idea is that FP arithmetic anyway suffers from numerical errors associated with rounding. Therefore, the author propose that FP arithmetic can be replaced altogether by pre-aligned integer arithmetic. Essentially, the representation is kept in FP, which preserves accuracy, but compute is done in integer, which is more efficient.

**Summary Of The Review:**

The paper introduces a new approximate way of computing FP MatMuls using integer arithmetic. The idea is novel and interesting. I have a few questions I hope the authors can answer.

---

> ### Author Response · Authors · 2022-11-15
> **Response to Reviewer 7bGi (1/2)**
>
> Thanks for the constructive comment.
>
> We will now discuss 1) hardware evaluation with memory access, 2) custom CUDA kernel for FP to INT MatMul emulation (response to the 3rd question), and 3) strength of the proposed work compared to the fully-integer baseline.
>
> **Response 7bGi-1. Hardware evaluation with memory access**
>
> As the reviewer suggested, we evaluated MatMul accelerators by accounting memory access for in-depth analysis of the proposed scheme in the real computing scenario. We updated the experimental results in Appendix C.5 of the revised draft. In summary, for energy efficiency, the amount of improvement is slightly decreased, but the proposed iFPU still improves the energy efficiency by 3.0 times to 6.6 times compared to the FP-MAC baseline. The throughput-per-area gain of the proposed iFPU over the baselines remains the same even when the memory access is considered.
>
> As the reviewer correctly pointed out, the cost of off-chip DRAM access is much more expensive than the computational cost. However, the MatMul of DNNs often reuses both activations and weights intensively, so the cost of DRAM access can be amortized as (DRAM access cost) / (number of data reuse). For example, when the sequence length of the BERT-base model is 128 and the batch size is 1, the feed forward layer with 768 x 3072 input/output dimension reuses input activation and weight 3072 (=output dimension) and 128 (=sequence length * batch size) times, respectively. In this case, the cost of DRAM access can be amortized by 3072 computations. For this reason, the overhead of DRAM access is not significant in the conventional MatMul acceleration system, and the proposed iFPU can achieve noticeable performance gain even in the real computing scenario in which memory access is considered. Moreover, because double buffering can overlap memory access with computation, the delay of DRAM access is perfectly hidden in the MatMul engines. Therefore, the throughput-per-area gain of the proposed iFPU can remain the same even when the memory access is considered.
>
> **Response 7bGi-2. Custom CUDA kernel for FP to INT MatMul emulation (response to 3rd question)**
>
> Table 7bGi-A. Comparison of execution time between PyTorch and our custom CUDA kernel
> | Operation Type || FP MatMul || INT-based FP MatMul ||
> |---|----|:---:|----|:---:|:---:|
> | Kernel || PyTorch CUDA backend || PyTorch CUDA backend | custom CUDA kernel |
> |Execution time ($\mu$s) || 11.0 || 14,438.0 | 77.1 |
>
> In our work, we designed and used a custom CUDA kernel to precisely emulate the FP to INT MatMul of the proposed iFPU. We deeply thank the reviewer for asking the emulation setup because implementing the custom CUDA kernel was a crucial enabler for us to continue this research due to the time consuming issues with conventional CUDA kernel.
>
> As the reviewer correctly pointed out, the emulation of the FP to INT MatMul on GPU can be much slower than conventional FP MatMul if it stores the intermediate results in memory. Hence, to speed up the emulation, we designed a custom kernel to keep the intermediate results in the local register instead. As a result, the custom kernel produces results for the emulation of the proposed INT-based FP MatMul much faster than the conventional CUDA kernel does. As shown in Table 7bGi-A, we could observe the orders-of-magnitude reduction in execution time when we used our custom CUDA kernel compared to PyTorch. For the measurements, we used a single FC layer inference with batch size = 200, the size of input = 256, and the size of output = 128 on RTX3090. For the emulation of the INT-based FP MatMul, the emulation takes 77.1 $\mu$s with custom CUDA kernel,  while it takes 14.438 ms with conventional PyTorch Operators. The custom CUDA kernel has 187x speeds up over PyTorch.
>
> Of course, emulation of INT-based FP MatMul with CUDA kernel is still slower than conventional FP MatMul with PyTorch (11 $\mu$s vs. 77 $\mu$s). But, the custom CUDA kernel made it possible for us to run various experiments in reasonable time, which was not possible with PyTorch.
>
> Please also note that such a slowdown occurs in GPU settings only. The proposed INT-based FP MatMul runs fast on the dedicated hardware (iFPU).
>
> For more details, we attached the custom CUDA kernel developed for the emulation in the supplementary material.

---

> > ### Author Response · Authors · 2022-11-15
> > **Response to Reviewer 7bGi (2/2)**
> >
> > **Response 7bGi-3. Strength of the proposed work compared to the fully-integer baseline (response to 2nd question)**
> >
> > As the reviewer mentioned, the fully-integer model can achieve high computational efficiency as it can naturally use integer arithmetic. However, although many previous works proposed methods to improve the accuracy of the fully-integer model, it is still challenging to preserve DNN accuracy after quantizing both weights and activations in many cases. For example, TPUv4i [2] states that it employs FP arithmetic units for inference because the accuracy degradation of some models are noticeable and it could take “months of extra development to restore the quality score using integers that experts achieved in floating point during training”.
> >
> > The paper [1] introduced by the reviewer successfully quantizes ResNet models without accuracy loss, but it encounters challenges in quantizing MobileNet models. As short retraining Quantization-Aware-Training (QAT) shows 2.39% accuracy degradation after the quantization, MobileNet-V2 requires long QAT to maintain the network accuracy as FP models. Moreover, for MobileNet-V3-small, quantization leads to around 8% accuracy degradation even with long retraining QAT combined with dynamic-optimal clipping. On the other hand, BRECQ [3] reported that 4-bit weight-only quantization (with FP activation) using simple Post-Training-Quantization (PTQ) can preserve the accuracy for a variety of DNN models such as ResNet, MobileNet, RegNet, and MnasNet. Moreover, weight-only quantization is also gaining attention as a solution for the NLP model compression as activation quantization is challenging for NLP tasks. Hence, we believe that the weight-only quantization can be a more general solution for model compression compared to the fully-integer models.
> >
> > While the fully-integer models replace FP arithmetic with integer arithmetic, the weight-only quantization still requires FP arithmetic for the DNN computation as activations remain FP values. For this reason, it is challenging to improve computational efficiency of weight-only quantized models on the conventional processors. Therefore, we proposed the iFPU to improve the computational efficiency of weight-only-quantized models by replacing FP-based MatMul to integer addition with proposed schemes such as pre-alignment. As shown in the experimental results summarized in Figure. 9 of the manuscript, computation of weight-only quantized model using conventional FP arithmetic logics (FP-MAC and FP-ADD engine) is much less efficient than computation of fully-integer model (INT8 engine). However, iFPU greatly improved the computational efficiency of weight-only quantized models, making the efficiency of iFPU comparable to that of the INT8 engine. Hence, with the proposed iFPU, we can win the advantage of FP activation in the perspective of DNN accuracy and the computational efficiency of integer arithmetic.
> >
> > Thanks again for the comments.
> >
> > [1] Sakr, Charbel, et al. "Optimal Clipping and Magnitude-aware Differentiation for Improved Quantization-aware Training." International Conference on Machine Learning. PMLR, 2022.
> >
> > [2] Norman P Jouppi, et al. “Ten lessons from three generations shaped google’s tpuv4i: Industrial product.” In 2021 ACM/IEEE 48th Annual International Symposium on Computer Architecture (ISCA), pp. 1–14. IEEE, 2021.
> >
> > [3] Yuhang Li, et al. “Brecq: Pushing the limit of post-training quantization by block reconstruction.” In International Conference on Learning Representations, 2021.

---

### Official Review · Reviewer_zxxg · 2022-10-28

**Confidence:** 4
**Correctness:** 2
**Technical Novelty And Significance:** 2
**Empirical Novelty And Significance:** 3
**Recommendation:** 5

**Clarity, Quality, Novelty And Reproducibility:**

Stating the size of the adder and multiplier blocks used in the PE and Scale and Accumulator blocks will be helpful to project the scalability to different quantization widths. Kindly also add figures showing the internal circuitry of these blocks to help improve the understanding of their design, similar to the pre-alignment unit.

Questions regarding the novelty and reproducibility (to larger quantized bit-size networks) are listed in Strengths and weakness section.



**Strength And Weaknesses:**

The paper presents a good overview of the target application by highlighting the limitations of the floating-point format and prior approaches. The diagrams presented are clear and help explain the proposed ideas. The authors derive the conclusion in Remark1 with detail which helps establish the important point that iFPU maintains the same error at the output despite lowering complexity of the operations performed compared to FP. The presented experiments are useful and highlight the capabilities of the proposed hardware implementation, iFPU as opposed to prior art.

While the authors present their ideas on truncation for FP multiplication, the idea of truncation in binary multipliers is not new, e.g., N. Petra, et. al., "Truncated Binary Multipliers with Variable Correction and Minimum Mean Square Error," in IEEE Transactions on Circuits and Systems I: Regular Papers, 2010, presents similar ideas which discuss truncating the bits from the partial products derived after binary multiplication and their impact on the output accuracy. In fact, the authors in Petra et. al., paper define an additional correction function estimated based on the truncated bits which further helps reduce the error measured. I request the authors to cite this paper and elaborate how their contributions are unique from those in that paper.

An important assumption in the proposed technique is the ability to simplify the quantized weight values to respective bit-planes, which allow simplifying the multiply ops to add/subs. For the paper the authors assume the use of 4-bit quantized weights which might favor the simplification of weights into +1/-1 easily. But often it is harder to achieve the same level of accuracy for a given ML task with a DNN with lower quantization bit sizes. As the quantization bit size increases to 8/16bit integer values, how do the benefits of the proposed iFPU scale? Since for larger bit size the number of bit-planes will increase which will increase the cost of the Scale and Accumulator block shown in Figure 5.

FP-Add is a useful baseline added by the authors which highlights the individual contribution of the bit-plan decomposition and truncation ideas used. I am curious why authors select Int8 Matmul engine considering that the iFPU assumes 4-bit quantized weights, it would be helpful to add an Int4 Matmul engine data as well cause that would be the baseline of the energy efficiency achievable.


**Summary Of The Paper:**

The paper presents a new method for the hardware implementation of deep neural networks (DNNs) which are quantized but use floating-point (FP) activations. Prior art has attempted to solve this problem by devising new data formats e.g., bfloat16 and Block Floating Point number for weights and activations, but they have limitations (high overhead to overcome rounding errors and lack of configurability for given hardware platform respectively).

The paper presents an alternative approach which optimizes the matrix multiplication (MatMul) operations using a special design block, iFPU. It involves decomposing quantized weights into bit-planes (taking out common factors and reducing weights into a series of +1/-1 values). Activation values (FP/bfloat16 format) are similarly reduced by normalizing them to the largest common exponent and converting the mantissa bits to integers. Thereafter, in this reduced space the multiply ops are converted to a series of integer adds (subs) which helps reduce the Matmul complexity.

 Further, the authors further simplify the add/sub ops by truncating the least significant bits of the integer values added upto the precision of the output plus a few bits. This truncation helps reduce the complexity of integer add/sub operations significantly. Once the add operations are done, the resulting values are converted back to FP format followed by scaling with values in each bit-plane (removed from weights) and a floating point add.

The authors demonstrate through detailed exploration of FP add operations that their proposed truncation technique doesn’t cause any error higher than the rounding error of existing FP add operations. Additionally, they also demonstrate that this continues to hold when testing on real networks by showing a strong match between the activation values calculated (in network layers) and comparing the output accuracy generated by FP and iFPU HW implementation.

Lastly, authors compare their proposed iFPU implementation with two other HW implementations, FP (default) and FP-ADD, which does similar decompose and add/sub in bit-plane space steps as iFPU without any truncation. The results show that iFPU achieves higher energy efficiency and area efficiency than the other implementations.


**Summary Of The Review:**

The paper presents strong results showing the accuracy and energy/area efficiency of the proposed HW implementation of FP activation and 4-bit integer quantized weights Matmul operations. But its not obvious that the benefits observed will scale for other quantization bit-widths. Further, as shown in Figure 9 a significant part of the energy/area efficiency benefits are derived due to truncation of FP add operations which relies on similar experiments as another paper referenced above. If that is removed from the results, the improvement in energy/area efficiency due to the other ideas presented in the paper are smaller.

---

> ### Author Response · Authors · 2022-11-15
> **Response to Reviewer zxxg (1/2)**
>
> Thanks for the constructive comment.
>
> We will now discuss 1) the differences between [Petra et al.] and the proposed method, 2) the benefits of the proposed iFPU with higher-bit weights, 3) comparison with INT8/INT4 MatMul engines, and 4) a detailed block diagram of iFPU and its bit-scalability.
>
> **Response zxxg-1: The differences between [Petra et al.] and the proposed method**
>
> As the reviewer rightfully suggested, the idea of truncation in binary multipliers is not new, and both of [Petra et al.] and our work discuss the truncation to improve computational efficiency.
> However, there are critical differences between the two works. First of all, [Petra et al.] dealt with integer multiplications while our paper focuses on floating-point (FP) additions/subtractions. Due to the differences in the number format (integer vs. FP) and arithmetic operations (multiplications vs. additions/subtractions), the two works present completely different error analysis models and error reduction schemes.
>
> The error analysis models between [Petra et al.] and our work are different because the amount of truncation is fixed in the truncated binary integer multipliers and the amount of truncation varies in our work as the amount of significand shift varies depending on the input data. Moreover, in truncated binary multipliers, the bit resolution of truncated output is defined by the application requirement. On the other hand, as we proposed to truncate the pre-aligned values to adopt lower-bit integers and improve computational efficiency, the proper bit resolution of truncated values should be found to meet the accuracy requirement in our case.
>
> In addition, in the integer multiplication case, some of the truncated partial products share the same inputs with the remaining partial products, so they have correlations with the remaining partial product values. [Petra et al.] proposed an error minimization scheme which exploits such characteristics. On the other hand, in the FP addition/subtraction case, the truncated significands do not have any correlation with the remaining bits so it is hard to devise similar error correction schemes. Instead, we focused on the fact that conventional FP operation is also not precise due to the rounding of output significands so that we only need to match the error level of the proposed scheme to that of the conventional FP operations. Based on the facts, we showed a theoretical analysis such that the proposed integer-based FP addition/subtraction can have the similar error level as that of the conventional FP addition/subtraction when a small number (1-2) of extra bits are attached to the shifted significands. With this finding, we can design an efficient integer-based FP addition logic without having complex error correction function estimated based on the truncated bits.
>
> As the reviewer suggested, we added [Petra et al.] to the reference list and left a brief comparison in Section 2.2. We also discussed the detailed comparison between [Petra et al.] and the proposed work in Appendix B.2.
>
> [1] Nicola Petra, et. al. “Truncated binary multipliers with variable correction and minimum mean square error.” IEEE Transactions on Circuits and Systems I: Regular Papers, 57(6):1312–1325, 2009.
>
>
> **Response zxxg-2: The benefits of the proposed iFPU with higher-bit weights**
>
> As the reviewer pointed out, the size of weight bits affects the performance gain of the proposed iFPUs. We updated the detailed analysis with various weight bits in Appendix C.3.
>
> Because the proposed scheme processes each bitplane of the weights in the bit-parallel manner, higher-bit weights require more operations with PE, scale, and accumulators. Hence, the benefits of the iFPUs diminish as the number of weight bits increases. Nevertheless, even for 8-bit weight case, iFPUs achieve better computational efficiency compared to the baseline as shown in Figure 17 of Appendix C.3.  Although 8-bit weight case in our design still shows benefits in terms of TOPS/W and TOPS/$mm^2$, we believe that lower bit ($\leq$ 4bit) weight with FP activation is a more promising design point. Several recent publications have presented results that making low bit ($\leq$ 4bit) weight quantization with excellent accuracy becomes easier when the activation is kept as the FP number than the case when both weight and activation values are quantized. So, we believe that $\leq$ 4bit weight / FP activation combination is very promising. For the recent publications regarding quantized weight and FP activation, please refer to our response to Reviewer yB16 (Response yB16-2. Impact of the weight quantization on DNN accuracy).

---

> > ### Author Response · Authors · 2022-11-15
> > **Response to Reviewer zxxg (2/2)**
> >
> > **Response zxxg-3: Comparison with INT8/INT4 MatMul engines**
> >
> > We used the INT8 Matmul engine as a baseline for comparison because it is one of the most popular types that are currently used for inference-target NPUs both in industry and academia. We wanted to see whether our engine for 4-bit weight / FP activation has comparable efficiency with the INT8 engine in terms of TOPS/W and TOPS/$mm^2$ while using FP activations. Based on the evaluation results, we believe that our engine which keeps the FP format for activation values is more promising to keep up with new complex models than INT8 engines.
> >
> > Per  reviewer’s request, we also added the data for INT4 MatMul engine in the Figure 18 of Appendix C.4 to provide more insight for the readers. As expected, INT4 MatMul shows higher energy efficiency. However, comparing our design with INT4 MatMul does not provide a good insight because quantizing both weight and activation to 4 bits may not provide desired accuracy in many cases.
> >
> > Several papers reported that most DNN models can be compressed by 4bit weight-only quantization, which compresses the weights to 4 bits with non-uniform quantization while using FP activations. However, multiplication between 4-bit weights and FP activations still requires FP operation, so it is hard to improve computational efficiency with weight-only quantization. Hence, the proposed iFPU is designed to improve the computational efficiency of weight-only quantized model by replacing FP operation with integer operation with proposed pre-alignment and truncation. Meanwhile, it is known that 4-bit quantization of both weight and activation tend to experience relatively large accuracy drops except some DNN models, so INT4 MatMul may not be promising for large models despite its high computational efficiency. Hence, we think that it is better to compare the proposed engine with INT8 engine instead of INT4 engine.
> >
> > **Response zxxg-4. A detailed block diagram of iFPU and its bit-scalability**
> >
> > We added Figure.14 in Appendix C.1 to show more details about the hardware blocks. The proposed iFPU is a bit-flexible accelerator which can handle variable bitwidth of weight values. The iFPU processes weights in bit-parallel manner by processing each weight bitplane in different columns of the PE array. For example, 4-bit weights use  4 PE columns for the computation, and 8-bit weights use 8 PE columns for the computation. After the integer-based summations are done in each column of the PE array, the integer results are converted into FP values and multiplied by scaling factors which represent the significance of each bitplane. Then, computing results of each bitplane are merged in the accumulator (FP adder) to finish the MatMul. As the output resolution of FP accumulation remains the same regardless of the size of the accumulation thanks to the characteristics of the FP format, the size of the accumulator does not need to increase for the increased weight bitwidth.
> >
> > Thanks again for the comments.

---

### Comment · Area_Chair_nKkP · 2022-12-13
**Connection with work of Mogami, 2020**

Could you comment on the relationship of your paper with

Deep Neural Network Training without Multiplications,
Tsuguo Mogami, 2020
https://arxiv.org/abs/2012.03458

(which seems to address the harder task of training, vs just inference?)

---

> ### Author Response · Authors · 2022-12-14
> **Response to "Connection with work of Mogami, 2020"**
>
> Thank you very much for introducing the valuable reference [1]. While [1], like our research, aims to replace floating-point (FP) arithmetic with efficient integer arithmetic, the two studies target different FP operations, and the approaches taken to replace them with integer operations are also quite different.
>
> ```
> for int_mul(const float a, const float b) {
>   int c = *(int*)&a + *(int*)&b - 0x3f800000;
>   return *(float*)&c;
> }
> ```
> Figure 1. Pseudo-code of the addition-to-int operation proposed in [1].
>
> **A summary of the differences between the two studies is as follows**
>
> **1.** [1] proposed a methodology to replace FP multiplication of DNN MatMul with integer addition (a.k.a. addition-to-int, described in Figure 1). Then, FP multiplication & FP accumulation of MatMul can be reconstructed as addition-to-int & FP accumulation, so [1] still requires FP additions for the MatMul. On the other hand, our work proposed a solution to replace FP accumulations with integer accumulations, and the entire MatMul can be processed with integer units.
>
> **2.** The addition-to-int [1] is an aggressive approximation of FP multiplication. The scheme accurately processes only the exponent term. On the other hand, it uses approximation for handling mantissa part. As a result, the difference between the addition-to-int operation result and the conventional FP multiplication result can be substantial.  Therefore, it is difficult to guarantee inference accuracy when the addition-to-int is directly applied to the inference of a pre-trained neural network. To address the issue, [1] proposed “addition-to-int”-aware training (Table 1 of [1]). On the other hand, the proposed iFPU in our work replaces the FP additions with integer additions and guarantees the same numerical computation accuracy as the conventional FP units. As a result, accurate DNN inference can be performed using the iFPU without any iFPU-aware learning.
>
> **Let us describe the limitation of the schemes proposed in [1] more in detail.**
>
> As shown in Figure 1, the addition-to-int first converts FP operands to the integer numbers which have the same bit-level representations as their FP counterpart, and these integer numbers are added instead of computing the FP multiplication. Because FP format packs sign, exponent, mantissa in order, the addition-to-int results of each data type is as follows (assuming that there is no overflow in the addition of each data type for simplicity):
>
> sign$_c$ = sign$_a$  + sing$_b$
>
> exponent$_c$ = exponent$_a$ + exponent$_b$ - 127
>
> mantissa$_c$ = mantissa$_a$ + mantissa$_b$
>
> As the exponents should be added in the conventional FP multiplication also, the addition-to-int computes the exponent terms correctly. In addition, if there is no overflow in exponent$_c$, the addition of sign values can replace XOR operation, so the addition-to-int also computes the sign term correctly. However, the two mantissas that are multiplied in the actual FP multiplication are added in the addition-to-int, and hence the final operation result becomes different from the actual FP multiplication.
>
> As the authors of [1] stated in the example on page 2 of [1], the correct result of FP multiplication of 1.5 x 1.5 is 2.25, but the addition-to-int produces 2 instead, resulting in an error of about 12.5%, which is not small in our opinion. Furthermore, the worst case error can be even larger than the example. The maximum error occurs when the mantissa$_a$ and mantissa$_b$ are maximum, that is, 2'b11... 11. In such a case, the addition-to-int operation result further deviates from the actual FP multiplication result. Therefore, it is difficult to maintain inference accuracy when addition-to-int is applied to the inference of the neural network which was trained using conventional FP operations. As a result, [1] adopted addition-to-int-aware training to use addition-to-int for DNN inference (‘addition-to-int’-aware training results are reported in Table 1 of [1]).
>
> It is also worthwhile to note that, while the addition-to-int simplifies the computation for forward path (or inference) by replacing the FP multiplication, it actually makes the gradient computation in the backward path more complicated. The equation for computing the gradient using the addition-to-int is ∂f(a,b)/∂a = $e'$($l$(a) + $l$(b))$l'$(a), derived in page 2 of [1]. To ease the computational burden, [1] also suggests an additional approximation (a-operation) instead of the exact operation (e-operation) for the gradient computation. Such an approximation in addition to the aforementioned inaccurate computational results of the ‘addition-to-int’-based FP multiplication incurs heavy burden in neural network training.
>
> On the other hand, our scheme does not require dedicated training caused by the approximation of FP operations.
>
> [1] Tsuguo Mogami. "Deep Neural Network Training without Multiplications." arXiv preprint arXiv:2012.03458 (2020).

---

### Decision · Program_Chairs · 2023-01-20

**Decision:**

Accept: poster

**Justification For Why Not Higher Score:**

Concerns remained on the level of novelty

**Justification For Why Not Lower Score:**

This can be bumped down, given the high bar for ICLR. Reviewers please ackn the feedback and get back to us

**Metareview: Summary, Strengths And Weaknesses:**

For neural network inference, the paper proposes to use integer arithmetic on (truncated) floating-point operands, specifically for already quantized neural network weights.
While reviewers liked the efficiency improvements, questions remained on the level of novelty, and it only remains narrowly on the accept side even after the author feedback and discussion.

We hope the authors will incorporate the several points mentioned by the reviewers in the final version.
We'd also encourage authors to cite some more of the related work, including e.g.

- Deep Neural Network Training without Multiplications,
  Tsuguo Mogami, 2020

which the authors have already commented on in the discussion on openreview, thanks for that

**Note From Pc:**

if the above contains the word "oral" or "spotlight" please see: "oral" presentation means -> notable-top-5% and "spotlight" means -> notable-top-25%. As stated in our emails, we are disassociating presentation type from AC recommendations

**Summary Of Ac-Reviewer Meeting:**

Discussions had to be done only here on open review, as we couldn't find a common time to meet as two reviewers couldn't reply and one email address was outdated.